# Inhibiting Infectious Bronchitis Virus PLpro Using Ubiquitin Variants

**DOI:** 10.3390/ijms26115254

**Published:** 2025-05-29

**Authors:** Vera J. E. van Vliet, Olivia Roscow, Kihun Kim, Brian L. Mark, Marjolein Kikkert, Christine Tait-Burkard

**Affiliations:** 1Molecular Virology Laboratory, Leiden University Center for Infectious Diseases (LUCID), Leiden University Medical Center, 2333 ZA Leiden, The Netherlands; 2The Roslin Institute, Royal (Dick) School of Veterinary Studies, University of Edinburgh, Easter Bush, Midlothian EH25 9RG, UK; 3Department of Biology, Faculty of Science, University of Waterloo, Waterloo, ON N2L 3G1, Canada; 4Department of Microbiology, University of Manitoba, Winnipeg, MB R3T 2N2, Canada

**Keywords:** coronaviruses, avian infectious bronchitis virus, antivirals, innate immunity, ubiquitin variants

## Abstract

Infectious bronchitis virus (IBV) is a coronavirus first isolated in the 1930s infecting chickens. IBV causes great economic losses to the global poultry industry, as it affects egg production and causes mortality by leaving the host susceptible to secondary bacterial infections. Even though vaccines are available, they are poorly cross-protective against new variants of the virus, which are always on the cusp of emerging. Effective antiviral therapies, or possibly the production of transgenic animals immune to IBV infection, are therefore sorely needed. As the papain-like protease (PLpro) of IBV has deubiquitinating activity besides its crucial ability to cleave the viral polyprotein, we have applied a novel strategy of selecting ubiquitin variants (UbVs) from a phage-displayed library that have high affinity to this viral protease. These UbVs were found to inhibit the deubiquitinating activity of PLpro and consequently obstruct the virus’s ability to evade the innate immune response in the host cell. By obstructing the proteolytic activity of PLpro, these UbVs were seemingly able to inhibit viral infection as assessed using immunofluorescence microscopy. Whilst virus infection was detected in around 5% of UbV-expressing cells, the virus was present in around 30–40% of GFP (control)-expressing cells. This suggests that the expression of UbVs indeed seems to inhibit IBV infection, making UbVs a potentially potent and innovative antiviral strategy in the quest for control of IBV infections.

## 1. Introduction

The avian infectious bronchitis virus (IBV) was first described in the USA in the 1930s and has since caused major economic losses to the global poultry industry [1,2]. Estimations of total losses in an infected flock vary but have been approximated to be up to USD 65,000 per week on a farm with around 1 million broilers [3]. IBV is a coronavirus predominantly infecting chickens but has also been found to infect other avian species such as turkeys and pheasants [4]. Infected birds mainly suffer from respiratory issues, but disease of the kidneys and gastrointestinal tract, as well as oviducts, is also reported for certain strains. Viral infection often leads to mortality of the birds due to secondary bacterial infections [1,3]. The infection of egg-laying chickens also has great economic consequences, as it causes a significant, and usually persistent, decline in egg production of the affected flock [5]. Poultry continues to be a major protein source for humans, but the available vaccines against IBV are poorly cross-reactive. There are currently no effective therapeutics against infection and disease, and it remains a threat to the global food supply and the health of this avian species.

IBV is a single-stranded positive-sense RNA virus in the genus *Gammacoronavirus*, order *Nidovirales*. Like other coronaviruses, it has a large genome of around 28 kb in size. Besides its structural and accessory genes coded for in its 3′ third, IBV encodes two large polyproteins in the 5′ two-thirds of its genome. These polyproteins are cleaved by viral proteases, encoded in the polyprotein sequence, to release mature non-structural proteins or nsps [6]. The papain-like protease encoded within nsp3, PLpro, releases the three N-terminal nsps, whilst the 3C-like or main protease (nsp5) cleaves the remainder and the majority of the cleavage sites in the polyproteins. This results in a total of 15 different non-structural protein subunits named nsp2-nsp16 [7]. Besides its ability to cleave the polyproteins, IBV PLpro also has deubiquitinating (DUB) activity, which is dependent on the same catalytic site and is thought to help the virus evade antiviral innate immune responses [8,9]. This DUB activity, first confirmed to be present among the coronaviruses for SARS-CoV PLpro, describes the protease’s ability to cleave ubiquitin chains from cellular substrates and subsequently affect downstream signaling of these substrates in innate immune pathways [10,11]. It was shown that expression of IBV PLpro has a negative effect on the production of interferon β (IFN-β) and that PLpro is able to deubiquitinate factors in this innate immune pathway, such as melanoma differentiation-associated protein 5 (MDA5) and TANK-binding kinase 1 (TBK1) [12,13].

This basic principle of the DUB activity of coronavirus PLpro enzymes can be exploited to produce inhibitor molecules, as PLpro is known to bind ubiquitin. Previously, our labs have identified variants of ubiquitin (UbVs) from a phage-display library of sequence variants that were effective at inhibiting the PLpros of MERS-CoV and SARS-CoV-2, as well as the turnip yellow mosaic virus protease, due to their strong binding to the proteases [14,15,16,17]. By this tight binding to PLpro, these UbVs were not only able to inhibit the DUB activity of the proteases but also directly inhibit PLpro’s proteolytic activity and thereby significantly reduce viral titers in infected cells. As IBV PLpro was found to have similar proteolytic activity to that of other coronaviruses [8], we aimed to develop UbVs that bind to IBV PLpro with high affinity and are able to inhibit the protease. We have utilized a phage-displayed library containing billions of ubiquitin sequence variants and performed repeated selections of these UbVs using a recombinant version of the papain-like proteases of the IBV laboratory-adapted Beaudette strain, as well as the M41 field strain of IBV.

These strains were selected because most research into IBV is based on the lab-adapted Beaudette strain, as it is one of the few that is able to successfully replicate in immortalized cell lines. However, the Beaudette strain is non-pathogenic to chickens and may not provide an accurate basis for the development of novel inhibitors. For this reason, the same strategy was applied to the IBV prototype “Mass” serotype strain M41, as this is a pathogenic strain of the virus that could provide a better opportunity to develop inhibitors that would be valuable to target real-world IBV infections. Recombinant PLpros derived from both strains were expressed in *E. coli*, affinity purified and used for UbV-selection. These UbVs were subsequently tested in cell-based assays against both viruses and their respective PLpro enzymes.

We were able to rapidly select UbVs with high affinity to IBV Beaudette PLpro, IBV M41 PLpro or both. Their tight binding to PLpro facilitated obstruction of the protease sites responsible for deubiquitination and immune evasion, as well as significantly inhibiting viral replication, showing the promise of such a protein product as a viral inhibitor for IBV. Here, we show that the UbV approach that was previously successful at targeting the human coronaviruses MERS-CoV and SARS-CoV-2 [14,15] is also a feasible approach for targeting relevant veterinary coronaviruses, such as IBV.

## 2. Results

### 2.1. IBV PLpro Has Deubiquitinating Activity

The underlying principle for the UbV-based antiviral strategy is the fact that viral PLpro enzymes bind ubiquitin (Ub) and have DUB activity. Therefore, it was imperative to confirm the DUB activities of IBV PLpro, of both the Beaudette and M41 strains. To accomplish this, the appropriate sequences for IBV PLpro from both strains were cloned into a mammalian expression vector. Their respective sequences are shown in Figure 1A, and expression was confirmed by immunofluorescence staining (Appendix A). Using this expression vector, the ability of IBV PLpro to deubiquitinate host substrates was assessed. DF-1 cells, which are spontaneously immortalized chicken embryonic fibroblast cells, were transfected with constructs expressing FLAG-tagged, wildtype Ub only or co-transfected to add V5-tagged IBV PLpro from either Beaudette or M41. When these lysates were analyzed by Western blotting, the Ub-only sample showed a dark smear, representing a multitude of ubiquitinated host substrates. This smear was clearly fainter when a construct expressing either IBV PLpro from Beaudette or M41 was co-transfected (Figure 1B), confirming that IBV PLpro from both strains reduced the amount of ubiquitin-conjugated host proteins. This reduction in the ubiquitin smear was not present when a catalytic mutant (“C”) of the proteases was expressed instead. This confirmed that the IBV PLpros from these strains have DUB activity [8,9] and that this activity is dependent on its catalytic motif. This showed promise that the basis for our proposed strategy of developing UbV inhibitors for IBV is valid.

### 2.2. IBV PLpro Affects Innate Immune Signaling

Since ubiquitination of cellular substrates is often regarded as a necessary post-translational modification for activation of the innate immune response, it was assessed whether IBV PLpro was able to inhibit innate immune signaling. A reporter plasmid containing a firefly luciferase-coding region under control of a chicken IFN-β (chIFN) promoter was used to measure IFN-β promoter activation. Transfections in DF-1 cells were performed to show that the chIFN-β promoter activity is upregulated upon co-transfection of poly I:C, a synthetic double-stranded RNA (dsRNA) analog (Figure 2). A plasmid encoding Renilla luciferase was co-transfected for all conditions as a transfection control and used to normalize corresponding firefly luciferase values within the same well. It was shown that upon co-transfection of plasmids encoding the PLpro enzymes of the IBV Beaudette (Figure 2A) or M41 (Figure 2B) strains, the IFN-β promoter activity was markedly inhibited, suggesting that IBV PLpro expression from both strains negatively impacts innate immune signaling. This reduction was not observed when their respective catalytic mutant “C” was expressed instead, showing that this immune evasive activity is dependent on the catalytic activity of PLpro.

### 2.3. Selection of UbVs with Affinity for IBV PLpro Beaudette and/or M41

Since PLpro from both the Beaudette as well as the M41 stains of IBV had a negative effect on the activity of the interferon promoter and have deubiquitinating activity, these proteases were deemed appropriate for selection of ubiquitin variant inhibitory proteins. In order to perform the ubiquitin variant library screen, IBV PLpros from the Beaudette and the M41 strains were expressed in *E. coli* and affinity purified. The His-tagged PLpro proteins were then used in a UbV phage display library screen [17]. Relative to wildtype Ub, the UbVs in the library were soft randomized in three regions of their amino acid sequence, annotated region 1 (aa 2–14), region 2 (aa 42–49) and region 3 (aa 62–76), relative to wildtype ubiquitin. By repeated screening, eight unique UbVs were selected (Table 1), which showed promising binding affinity to one or both IBV PLpro proteins. The number of amino acid substitutions in these UbVs, compared to wildtype ubiquitin, ranged from three to sixteen, corresponding to a 3 to 20% sequence divergence compared to wildtype ubiquitin.

As explained and shown in Table 1, amino acid substitutions were restricted to three main regions by design, with some UbVs mutated in all three regions and others only in regions 1 and/or 3. UbVs 3 and 6 emerged from the screen and these contained quadruple glycines (G) at their C-termini, as the library was designed to contain slightly extended ubiquitin molecules (78 amino acids long). As wildtype Ub consists of only 76 amino acids, ending in a double glycine, two versions of these UbVs were used for further testing, one carrying a C-terminal quadruple G (UbVs 3 and 6) and one ending with a double G (UbVs 4 and 7), as it was unclear whether this distinction would affect their binding capabilities. In addition, special attention to the substrate conjugation activity of these particular UbVs was given, as wildtype Ub is conjugated to cellular substrates with its C-terminal glycine, which these UbVs also possess, meaning that they could be attached to various cellular substrates in the same way as wildtype Ub and thereby cause off-target effects. This concern was not as prevalent for the remaining UbVs, as these do not have C-terminal glycine residues. From this original screen, UbVs 1–7 were isolated from selections against IBV Beaudette PLpro and demonstrated high affinity in crude ELISA tests. Only UbV 8 emerged from the selection against IBV M41 PLpro, for which it also demonstrated high affinity. Nonetheless, the potential inhibitory activity of all UbVs was assessed using both PLpro enzymes.

### 2.4. UbVs Inhibit the DUB Activity of IBV PLpro

The selected UbV-coding sequences were cloned into expression vectors, so that their activity against IBV PLpro could be assessed in cell-based assays first. Expression was confirmed by immunofluorescence staining, as it was not possible to observe the UbV signal on Western blots (Appendix A). The DUB-inhibiting properties of the UbVs were tested using a similar assay as described above. Together with plasmids encoding wildtype Ub and either IBV Beaudette PLpro (Figure 3A) or IBV M41 PLpro (Figure 3B), the plasmids encoding different UbVs were co-transfected in DF-1 cells, and lysates were subjected to Western blotting analysis. This revealed that multiple UbVs are able to inhibit the DUB activity of IBV Beaudette PLpro in this assay, with UbVs 1, 2, 5, 6, 7 and 8 seemingly being the most active, as the smears of Ub-conjugated cellular proteins in these lanes were more prominent than in the PLpro-only lane (Figure 3A, compare lane 2 to lanes 4–11). There was also a clear inhibitory effect of these UbVs on the DUB activity of IBV M41 PLpro for the same reason. For this particular protease, UbVs 1, 2, 5, 7 and 8 yielded the strongest inhibition of DUB activity (Figure 3B, compare lane 2 to lanes 4–11).

As mentioned previously, some of the UbVs contained free glycine residues at their C-terminus, just like wildtype Ub, meaning that they could potentially be conjugated to cellular substrates in the same way as wildtype Ub. Expression of the UbVs and subsequent Western blotting revealed that UbVs 6 and 7 can indeed be conjugated to host proteins and could therefore have unwanted side effects (Appendix A). These UbVs were therefore not further explored.

### 2.5. UbVs Mitigate the Immune Evasive Activity of PLpro

As IBV PLpro from both the Beaudette and M41 strains negatively affected the IFN-β promoter activity (Figure 2), it was tested whether the UbVs were able to mitigate this effect and restore promoter activity. To accomplish this, a luciferase assay similar to Figure 2 was performed, this time co-expressing the UbVs. All UbVs were separately tested against IBV PLpro from the Beaudette strain and M41 strain. The luciferase assay performed with IBV PLpro from Beaudette (Figure 4A) showed that most UbVs, except UbV 3, weaken the inhibition of the IFN-β promoter activity by PLpro. Amongst those UbVs, UbV 1, 2, 5 and 8 had the strongest effects, in some cases bringing the activity of the promoter back to 100%. This suggests that these UbVs are able to mitigate the detrimental effect of IBV PLpro from the Beaudette strain on an important part of the innate immune system of the host cell.

All UbVs were also tested for inhibition of IBV M41 PLpro activity (Figure 4B). As was found for IBV Beaudette PLpro, co-expression of most of the UbVs caused a reduction in the inhibition of the IFN-β promoter activity compared to expression of only IBV M41 PLpro. UbV 6 and its companion UbV 7, which is two glycines shorter, showed no effect in this assay. Again, UbVs 1, 2, 5 and 8 are the most effective at inhibiting both IBV Beaudette and M41 PLpro. This shows that PLpro from both strains of IBV can be inhibited by the same UbVs, possibly indicating a pan-strain efficacy. This inhibition corroborates the previous observation in which these UbVs demonstrated inhibition of the deubiquitinating activities of both PLpros, as assessed in Figure 3. UbVs 1, 2, 5 and 8 proved to be the most effective at inhibiting IBV PLpro in these cell-based assays. The dose dependency of the UbV-driven inhibition of DUB activity was then verified by testing different doses of UbV 1, as they showed promising activity against both Beaudette and M41 PLpros, which confirmed the increased inhibition of respective PLpro enzymes with higher doses (Figure 4C,D).

### 2.6. UbVs Inhibit DUB Activity of PLpro In Vitro

In order to determine the degree to which the best-performing UbVs in the cell-based assays inhibited the DUB activity of the PLpro enzymes in vitro, we prepared recombinant protein in *E. coli*. The DUB activities of recombinant proteases of both the Beaudette and the M41 strains of IBV were measured in vitro using 7-amino-4-methylcoumarin-conjugated ubiquitin (Ub-AMC) as a substrate. They both showed similar enzyme kinetics (Figure 5A), where PLpro of M41 showed slightly lower V_max_ compared to Beaudette, which correlates with the slightly lower specificity constant of M41 PLpro for Ub-AMC as a substrate.

Among the various substitutions of UbVs relative to wildtype ubiquitin (Table 1), substitutions on the 10th amino acid from the N-terminus to Alanine or Valine are known to result in a dimeric state of UbVs [18]. Between the selected UbVs (UbVs 1, 2, 5 and 8), UbV 1 has a Glycine to Alanine (G10A) substitution. Therefore, as expected, UbV 1 showed two peaks with different retention times in its size exclusion chromatography elution profile (Appendix A). To compare the behavior of a UbV 1 monomer versus the dimer, these two states of UbV 1 were purified separately and used for the inhibition assays. Including the monomeric and dimeric states of UbV 1, all UbVs showed concentration-dependent inhibition of PLpro, with a nanomolar range of IC_50_ values (Figure 5B). The tested UbVs inhibit the DUB activity of PLpro significantly better than wildtype Ub, as Ub did not affect PLpro enzyme activity in the sub-micromolar concentration range tested. Both Beaudette and M41 PLpro enzymes showed a similar extent of inhibition by UbVs, with UbV 5 as the strongest inhibitor, and UbV 8 being the most unlikely candidate to show a significant effect on PLpro among the tested UbVs (Figure 5C). These in vitro DUB activity inhibition assays largely agree with the results from the cell-based experiments (Figure 3 and Figure 4), although in these previous experiments UbV 5 was not as potent as in these in vitro experiments. This could be due to the lower sensitivity range of the cell-based assays or the different context in which these experiments were performed.

### 2.7. Structural Basis for the Inhibition of PLpro by UbVs

To inspect the binding between the UbVs and the PLpro enzymes, crystallization of the complex of UbVs and PLpro was attempted. Despite several attempts with different conditions, crystallization trials failed to yield any crystals. To model the complex, AlphaFold2-Multimer [19] was utilized, and it predicted structures in the same binding mode as wildtype Ub but with unreliable scores. This binding prediction is due to the high sequence similarity of the UbVs and wildtype Ub. The low ipTM scores of binding interfaces are likely due to the UbVs having elongated bulky side chains at their C-terminus compared to wildtype Ub, which may affect its interaction with the PLpro active site in ways that are difficult to predict.

According to the previously reported X-ray crystal structure of the complex between wildtype Ub and IBV Beaudette PLpro (PDB ID: 5BZ0), all three domains of PLpro interact with Ub (Figure 6A) [8]. The binding interface area (1221.8 Å^2^ by PISA) [20] covers a large portion of Ub surface area (4516 Å^2^), including regions 1, 2 and 3. Thus, this may be considered the preferred binding mode of Ub with the largest binding interface area. Without the actual structure of the UbV-PLpro complex, we cautiously speculate that substitutions in UbVs might contribute to their tight binding, assuming that they adopt the same binding mode as wildtype Ub, which has been observed for other UbV-DUB complexes [14].

Region 1 of Ub, which includes a flexible loop between the β1 and β2 strands, forms crucial interactions with PLpro (Figure 6B). However, substitutions of G10 with hydrophilic amino acids may lead to additional interaction, as sidechains can protrude into the cleft formed by hydrophilic amino acids of PLpro (R200, N210, S212). G10R substitutions of UbVs 5, 6, 7 and 8 may fit in this same cleft. Region 2, which includes an I44 hydrophobic patch, does not interact with PLpro via hydrophobic interaction, as its counterpart interface on PLpro is hydrophilic (Figure 6C). R42 and N49 are located within 4 Å from PLpro site W156, though they do not interact. Substitutions in this region will change the surface shape and charge which may affect the binding. R42G in UbVs 1, 2, 3 and 4 and the removal of the hydrophilic bulky sidechain could aid W156 of PLpro in adopting an energetically proper conformation.

Region 3, which includes the C-terminal tail of wildtype Ub, extensively interacts with the area near the active site (C101) of PLpro (Figure 6D). The C-terminal tail (amino acids 70–76) of Ub mainly interacts with PLpro by its backbone amides, so substitutions in this region are unlikely to affect binding. However, the sidechain of L73 of Ub interacts with the hydrophobic interface of PLpro and is strictly conserved in all UbVs. G75 and G76 are surrounded by the PLpro BL2 loop; thus, substitution with a bulky sidechain may disrupt binding with the active site. In this case, the C-terminal tail of the UbVs may interact with a totally different interface of PLpro. One possible interface may be the BL2 groove between the BL1 and BL2 loops with hydrophobic residues (Figure 6D), as this groove is reported to be responsible for inhibitor binding for SARS-CoV PLpro [21,22,23]. In the case of UbVs 1, 2, 5 and 8, substitution of G75 and/or G76 may hinder the C-terminus to the active site regulated by the BL2 loop, and its actual binding mode should be investigated further.

### 2.8. UbVs Inhibit IBV Beaudette and M41 Infection

As the inhibiting effect of the UbVs on the IBV PLpro DUB activity and its immune evasive activities are now established, it was tested whether the UbVs are effective in inhibiting viral IBV infection and replication. Infections with IBV Beaudette and IBV M41 were performed at a multiplicity of infection (MOI) of 0.5 on DF-1 cells transfected with the plasmids encoding the four most promising UbVs (1, 2, 5 and 8) or GFP as a control. After 24 h post-infection (hpi), the supernatant from the Beaudette-infected cells was harvested, and the production of infectious viral progeny was measured by plaque assay. IBV M41 did not form plaques in our experimental set-up and was thus excluded from this assay. A slight decrease in the IBV Beaudette titer was found upon the expression of some of the UbVs in the cells compared to the GFP control, with UbV 5 and 8-expressing cells showing a 2-fold decrease in viral titers compared to the GFP control cells, indicating a possible antiviral effect of these UbVs (Figure 7A).

As mentioned, only a slight decrease in IBV titers was found in UbV 5 and 8-expressing cells. This is likely due to the low transfection efficiency of these cells (5–10%), which leaves many susceptible to infection and available to produce normal virus titers, rendering the difference between UbV-expressing and GFP-expressing control cells too small to be measured using a plaque assay. For this reason, it was decided to look at the single-cell level at UbV (or GFP control) expression and whether these positively transfected cells are infectable or not, using an immunofluorescence assay (IFA). To accomplish this, coverslips of transfected DF-1 cells that were infected 24 hpt with either IBV Beaudette or M41 were fixed using 3% paraformaldehyde (PFA) at 24 hpi. Despite the absence of a large decrease in viral titer, the IFA results show little to no IBV Beaudette infection of UbV-expressing cells, while the GFP-expressing control cells were frequently infected (Figure 7B). A representative experiment using UbV 2 is shown in Figure 7D. Similar observations were made with UbV-expressing cells infected with IBV M41 (Figure 7C,E). Images for the remaining UbVs (1, 5 and 8) are shown in Appendix A.

Though only a slight difference in viral titer was found, most likely due to the set-up of the assay, immunofluorescence staining of UbV-transfected IBV-infected cells showed that those cells that express UbVs are clearly less likely to show productive infection than those transfected with GFP as a control. This suggests that in addition to inhibiting the DUB activity of PLpro and its detrimental effect on IFN-β promoter activation, the UbVs are likely to inhibit IBV infection and viral replication.

## 3. Discussion

To this day, the coronavirus avian infectious bronchitis virus (IBV) has a major detrimental impact on the global poultry industry [1]. The virus not only causes respiratory disease but also causes decreases in egg quality and production, as it affects the oviducts of chickens. Even though vaccines against this virus are available, they are not adequate at halting infection and/or spread for multiple reasons, as they are poorly cross-protective. Furthermore, the vaccines used are often live attenuated vaccines which have been reported to recombine with field strains, as well as exert immunological pressure, consequently stimulating the emergence of new variant viruses [24,25]. This in turn promotes the genetic diversity of IBV, making it even more difficult to produce effective vaccines.

Some progress has been made regarding antiviral agents. Recently, a host defense peptide, called chicken cathelicidins 2, was identified as having anti-IBV activity [26]. However, clear differences in effectivity between study models and IBV strains were found, calling into question the usefulness of such a peptide as a broadly effective antiviral treatment for multiple strains. Other antivirals that have previously been suggested to work against coronaviruses like SARS-CoV-2, such as ivermectin, were unfortunately ineffective at inhibiting IBV replication [27]. Besides these options, many researchers have assessed the usefulness of traditional Chinese medicine or other plant-based substances against IBV infection. Some of these, such as a combination of Shegandilong with doxycycline, or the use of plant extracts, were indeed found to inhibit IBV infection to a certain extent in vivo [28,29]. Furthermore, the natural product myricetin was found to have antiviral activity against IBV PLpro [30]. These have, however, not been approved for treatment or prophylaxis. New strategies for fighting and/or preventing IBV are therefore sorely needed.

The DUB activity of IBV PLpro has been previously established; however, the genome sequence of an IBV QX-strain was used, which is largely restricted to southern China and is quite divergent from the Massachusetts-like strains found globally [9,31]. The authors also claimed that PLpro domain itself had no DUB activity and that only when the nsp3 transmembrane domain downstream of the protease domain was included in their construct did they detect such activity. Accordingly, we investigated whether a similar condition applied to the PLpro domain of the Beaudette and M41 strains. As our strategy was highly dependent on the binding of PLpro to ubiquitin, and thus DUB activity, we deemed it imperative to confirm PLpro activity before UbV selection.

We found that the IBV PLpro domains from both the Beaudette and M41 strains each exhibit a clear DUB activity, which is not present when the catalytic mutant of the proteases is expressed instead. This suggests an important role for the catalytic cysteine in this activity, which is expected, as deubiquitination entails a proteolytic cleavage of the covalent ubiquitin–substrate isopeptide bond. Furthermore, we found that both PLpro enzymes have a detrimental effect on the innate immune response in the cell, as measured by a reduction in activity of the chicken IFN-β promoter in our assay. Even though similar results have been published before, they were obtained with the use of the human instead of the chicken IFN-β promoter, possibly yielding skewed results [32]. Others that did use the chicken IFN-β promoter only included IBV PLpro from the M41 strain in their experiments [12].

Having established both the DUB activity of the protease as well as its negative effect on innate immunity, it was deemed appropriate to screen for UbV binding. UbV selection using both Beaudette and M41 PLpros provided us with eight promising original UbVs (Table 1). When tested in a cell-based DUB assay, UbVs 1, 2, 5 and 8 were found to inhibit the DUB activity of either Beaudette PLpro, M41 PLpro or both (Figure 3). Assessing their effectiveness in the luciferase assay showed that expression of these same UbVs inhibited the detrimental effect of the PLpro enzymes on the innate immune response in the cell significantly (Figure 4). UbVs 1, 2, 5 and 8 were subsequently assessed in biochemical in vitro studies.

The recombinant PLpro enzymes from Beaudette and M41 showed similar Michaelis–Menten kinetics (Figure 5A), which was not unexpected given that their sequence differences do not map to active site or Ub binding interfaces (Figure 1). The k_cat_/K_M_ value of Beaudette PLpro (0.49 ± 0.027 µM^−1^s^−1^) was different from previous reports using an LRGG-AMC substrate (0.67 ± 0.11 mM^−1^s^−1^) [8], due to the significant differences between substrates. Ub-AMC is a full-length Ub conjugated to a fluorophore on its C-terminus, while the short peptide substrate used by Kong and co-workers only mimics the four C-terminal amino acids of Ub. Ub-AMC is arguably a better mimic of the natural interaction between PLpro and Ub. Furthermore, in our biochemical assay, all UbVs tested showed IC_50_ values in the nanomolar range (Figure 5C) compared to wildtype Ub, which did not show appreciable inhibition of PLpro DUB activity. For both Beaudette and M41 PLpros, UbV 5 showed the strongest inhibition (IC_50_ ~9.7 nM and ~4.4 nM, respectively). These low IC_50_ values of UbVs agree with our hypothesis that the UbVs selected by phage display inhibit PLpro DUB activity efficiently.

In contrast with the other tested UbVs, UbV 1 has a G10A substitution compared to ubiquitin, which leads to a conformational change that promotes dimerization, as well as a possibly increasing affinity for the target [16,18,33]. In previous studies, this substitution led to the exchange of a β1 strand between two UbV chains. Therefore, UbV 1 likely exists in both monomeric and dimeric conformations. Both UbV 1 conformations were purified, and their IC_50_ values were measured, which were comparable. This suggests that the conformational change caused by dimerization may not affect its binding mode to PLpro.

Previously, we established that UbVs are promising antiviral agents against MERS-CoV and SARS-CoV-2 PLpro [14,15]. In the current study, we extended this strategy to a veterinary coronavirus, IBV. The tested UbVs show promising inhibition of the various functionalities of IBV PLpro from two strains, as well as seemingly inhibiting viral infection, as assessed by immunofluorescence assay. The limited effect on infectious viral progeny titers that we observed in our assays was likely due to the transfection efficiency in the DF-1 cells being relatively low. This leaves many cells susceptible to infection and available to produce normal virus titers, rendering the difference with GFP-transfected control cells too small to be measured. For future research, another set-up of this experiment, for example using cell lines that stably express the UbVs, would likely yield more convincing results. Nonetheless, in vitro, the selected UbVs seemed similarly effective at inhibiting the PLpro enzymes from both IBV Beaudette and M41. Structural modeling revealed that the amino acid differences between Beaudette and M41 do not overlap with the binding interfaces of Ub and are thus not likely to interfere with UbV binding either. Thus, this UbV approach may be an efficient way to target a variety of IBV strains, holding promise for broad-acting anti-IBV effects.

In the future, it would be interesting to explore the possibility of producing genetically modified chickens that express one or more IBV-inhibiting UbVs, aiming to render the animals resistant to IBV infection. Indeed, others have been able to genetically edit chicken primordial germ cells to contain a mutated version of the protein ANP32A, which in turn prevents the interaction of the avian influenza viral polymerase with this host protein [34]. These germ cells were then used to produce a genetically modified chicken resistant to low-dose influenza infection. Further research is needed to determine whether UbVs are sufficiently efficacious as antiviral agents in whole organisms to treat and/or prevent coronavirus infections.

## 4. Materials and Methods

### 4.1. Cell and Virus Culture

Chicken embryonic fibroblasts (DF-1 cells) were maintained in Dulbecco’s modified Eagle’s medium (DMEM, Lonza, Basel, Switzerland), supplemented with 10% fetal bovine serum (FBS, Capricorn Scientific, Ebsdorfergrund, Germany), 1× non-essential amino acids (NEAA, Merck, Darmstadt, Germany), and 1× glutamine (Ala-Gln, Merck). The cell line tested negative for mycoplasma.

IBV strains Beaudette and M41 were propagated in fertilized chicken eggs or DF-1 cells and used for infection experiments inside BSL-2 facilities at the Roslin Institute and the Leiden University Medical Center. IBV M41 was a kind gift from Sjaak de Wit at Royal GD (Deventer, The Netherlands). The complete genome reference sequence of these strains is available under GenBank accession numbers NC_001451.1 (IBV Beaudette) and DQ834384.1 (IBV M41).

### 4.2. Antibodies

For Western blot analysis, the following antibodies and corresponding dilutions were used: mouse anti-FLAG (clone M2, #F1804, Sigma-Aldrich, Saint Louis, MO, USA, diluted 1:2000), mouse anti-HA (HA.C5, Genetex, Irvine, CA, USA, 1:1000) and mouse anti-α-Tubulin (clone B-5-1-2, #T5168, Sigma-Aldrich, diluted 1:2000). These primary antibodies were detected with biotin-conjugated goat anti-mouse IgG (#31802, Invitrogen, Waltham, MA, USA, diluted 1:2000) and tertiary antibody Cy3-conjugated mouse anti-biotin (#200-162-211, Jackson, Avon, CT, USA, diluted 1:2500).

For immunofluorescence assays, the following antibodies and dilutions were used: rabbit anti-FLAG (#F7425, Merck, diluted 1:500), mouse anti-dsRNA (clone J2, Scicons, Springville, CA, USA, diluted 1:1000) Alexa488-conjugated goat-anti-rabbit IgG (Invitrogen, diluted 1:300) and Cy3-conjugated donkey anti-mouse IgG (Jackson ImmunoResearch Laboratories, West Grove, PA, USA, diluted 1:1000). Nuclei were stained with Hoechst 33258 (Thermo Fisher Scientific, Waltham, MA, USA, diluted 1:100).

### 4.3. Plasmid Construction

Plasmids pcDNA3-V5-IBV-PLpro-Beaudette, pcDNA3-V5-IBV-PLpro-Beaudette-C, pcDNA3-V5-IBV-PLpro-M41 and pcDNA3-V5-IBV-PLpro-M41-C were constructed using synthetic gBlocks (IDT), cloned in-frame with a V5 tag on the N-terminal side into a pcDNA3 vector using the XhoI and KpnI restriction enzymes. The GenBank sequences NC_001451.1 (IBV Beaudette) and DQ834384.1 (IBV M41) were used to determine the sequences of PLpro at genome positions aa1348-1659 (Beau) and aa1350-1661 (M41). Catalytic mutants were produced by site-directed mutagenesis to create C101A mutations in both constructs. The pcDNA3-FLAG-UbV plasmids were cloned into the same vector using the NheI and BamHI restriction enzymes. Constructs were verified by Sanger sequencing. Other plasmids were described elsewhere or provided by others: pLenti6.3-FLAG-Ub.WT [14], pRK5-HA-Ub was a gift from Ted Dawson (Addgene plasmid #17608; http://n2t.net/addgene:17608, URL accessed on 13 April 2022; RRID:Addgene_17608 [35]) and pcDNA3.1-empty (Invitrogen). Sequence alignments were produced using Geneious version 10.2 created by Biomatters, New York, NY, USA.

For the luciferase assay, the pGL3-Basic Vector was used to clone the chicken IFN-β promoter region. This insert was ordered as gBlock (IDT), rehydrated and digested using NheI and HindIII restriction enzymes, as was the vector backbone. These were ligated using T4 DNA ligase (NEB, Ipswich, MA, USA).

For expression and purification, the sequences encoding the IBV PLpro protease domains of IBV Beaudette and IBV M41 were identified and ordered as gBlocks (IDT), as mentioned above, and ligated into the bacterial expression vector pETM11, which provides the inserted sequence with a His_6_-tag.

### 4.4. Expression and Purification of IBV PLpro for UbV Screening

His_6_-IBV PLpro enzymes were expressed from the pETM11-based vector in *E. coli* BL21 (DE3) cells (Stratagene). Transformed bacterial cells were grown in 2×YT medium containing kanamycin, overnight. The next day, the prewarmed Luria–Bertani (LB) medium was spiked with the overnight culture and supplemented with kanamycin. The cultures were incubated at 37 °C with shaking until the OD_600_ reached 0.7. Cultures were then induced with IPTG (final concentration 0.1 mM) and incubated overnight at 16 °C, 180 rpm. The overnight cultures were spun down, and cell pellets were lysed using bacterial lysis buffer (50 mM NaH_2_PO_4_, 300 mM NaCl, 30 mM Imidazole, 10% glycerol and cOmplete^TM^ Protease Inhibitor Cocktail (Roche, Basel, Switzerland)), supplemented with lysozyme, Triton X-100 and Benzonase. Lysates were incubated on a rotator for a few hours at 4 °C. HisPur Ni-NTA Resin slurry (Thermo Fisher Scientific) was then used for protein purification by batch method using the corresponding user guide. Finally, a buffer exchange was performed using a 10 kDa cut-off centrifugal filter tube to remove excess salts. Eluted protein was analyzed by SDS-PAGE (containing 2,2,2-trichloroethanol) and subsequently analyzed upon UV exposure to verify proper size and purity of the sample (Appendix A).

### 4.5. Selection of UbVs

UbV selection was performed as described by Zhang and Roscow [36]. In brief, purified IBV Beaudette and M41 PLpro were diluted in PBS to 1 µM and coated on hydrophobic microplates overnight at 4 °C. In preparation for the first round of phage display, a seed culture of *E. coli* OmniMAX was prepared by inoculating 2YT broth supplemented with tetracycline with an isolated colony and incubated overnight at 37 °C (assume all incubation steps are performed with shaking, unless stated otherwise). The next day, the seed culture was used to seed a mid-log phase OmniMAX culture. Unabsorbed protein was poured off, and the plate was blocked with 1% *w*/*v* bovine serum albumin (BSA)-supplemented PBS (PB buffer) by rinsing once, refilling and incubating at 4 °C for one hour. During this time, the phage library was thawed on ice and diluted 10-fold in PBS. One-fifth the volume of 20% polyethylene glycol/2.5 M NaCl (PEG-NaCl) was added and mixed before incubating on ice for 30 min. The solution was centrifuged at 11,000× *g* for 30 min at 4 °C, after which supernatant was discarded, and the phage pellet was gently resuspended in 1 mL of 0.05% Tween 20- and 1% *w*/*v* BSA-supplemented PBS (PBT buffer). Once the blocked plate was finished incubating, the blocking solution was poured off, the phage library was added, and the plate was incubated at room temperature for one hour to allow phage binding to recombinant PLpro. Unbound phages were then poured off, and wells containing bound phage were washed four times with 0.05% Tween 20-supplemented PBS (PT buffer). Phages were eluted by incubation with 0.1 M HCl at room temperature for 5 min. The pH was neutralized by adding 12.5% *v*/*v* 1 M Tris-HCl (pH 11) and pooled eluted phages were supplemented with 1% BSA and stored at 4 °C. This solution is termed the phage output. Half of the round one output volume was used to inoculate the mid-log phase cells prepared earlier, which were then incubated at 37 °C for 30 min. M13K07 helper phages were cultured to a final concentration of 10^10^ PFU/mL, and the culture was incubated at 37 °C for another hour. The culture was seeded into 2YT broth supplemented with carbenicillin and kanamycin and grown overnight at 37 °C. The next day, the culture was centrifuged at 11,000× *g* for 10 min at 4 °C, and the supernatant was decanted and mixed with PEG/NaCl. After incubating on ice for ten minutes, the solution was centrifuged at 11,000× *g* for ten minutes at 4 °C. The supernatant was discarded, and the phage pellet was resuspended in PBT, centrifuged at 16,200× *g* for four minutes at 4 °C to pellet debris and stored at 4 °C. This enriched library solution was used as input for the next round of selection. Five subsequent rounds of selection were performed in a similar manner, with each round receiving the enriched input from its respective previous round. Only the post-selection washing step becomes more stringent with each round of selection, to increase the likelihood of enriching a binder with significantly increased affinity for IBV PLpro. A mid-log phase *E. coli* OmniMAX culture was inoculated with ten-fold PBS dilutions of rounds four and five’s phage outputs and plated onto multiple LB plates supplemented with carbenicillin. Colonies were picked, inoculated into 2YT broth supplemented with carbenicillin and M13K07 helper phage and grown overnight at 37 °C. Cultures were centrifuged at 1200× *g* for ten minutes at 4 °C, and supernatant was transferred to a new container and stored at 4 °C. These solutions, containing eluted and amplified phage DNA, were then analyzed via Sanger sequencing. Sequencing data were cleaned and analyzed using an in-house program (PDAS: https://github.com/oroscow/phage_display_analysis_suite_pdas, URL accessed on 9 September 2023).

### 4.6. Luciferase Assays

DF-1 cells were seeded into 24-well plates and transfected when cell confluency was around 70%. Transfections were performed using pGL3-chIFNβ-luc, pRL-TK (Renilla) and different combinations of pcDNA3-V5-IBV-PLpro-Beaudette, pcDNA3-V5-IBV-PLpro-Beaudette-C, pcDNA3-V5-IBV-PLpro-M41, pcDNA3-V5-IBV-PLpro-M41-C and pcDNA3-FLAG-UbVs. pcDNA3.1-empty vector was used to normalize the total amount of transfected DNA per well. Poly I:C (HMW, Invivogen, San Diego, CA, USA) was used as an inducer of the chIFN-β promoter activity in this assay. Linear polyethylenimine (PEI 25K, Polysciences, Warrington, PA, USA) was used as a transfection method. Briefly, per 1 μg of plasmid DNA, 3 μL of PEI (1 μg/μL) was diluted in Opti-MEM medium (Lonza). The plasmid DNA was also diluted in Opti-MEM. Diluted PEI was then added to the diluted DNA, vortexed and incubated for 20 min. The mixture was then added to the cells (24-well plate, 5 × 10^4^ cells/well) in a dropwise manner, and the cells were incubated at 37 °C, 5% CO_2_. Cells transfected with the luciferase assay were lysed 18 h post-transfection in 1×PLB (Promega, Madison, WI, USA). Read-out was performed using the Dual-Luciferase Reporter Assay System (Promega), and the CLARIOstar Plus Microplate Reader (BMG Labtech, Ortenberg, Germany) or EnVision Multilabel Plate Reader (PerkinElmer, Waltham, MA, USA). Corresponding graphs were created using Graphpad Prism (version 9). All luciferase assay experiments were performed at least three times independently, with four technical replicates each. Unpaired two-tailed Student’s *t*-tests were performed to determine statistical significance.

### 4.7. Deubiquitination (DUB) Assays

Deubiquitination assays were performed as described previously [14]. For the current study, DF-1 cells were seeded into 12-well plates and transfected when cell confluency was around 70%. Combinations of tagged ubiquitin, PLpro and UbVs were expressed to assess the ability of PLpro to deubiquitinate (poly-) ubiquitinated proteins in the cells and whether the UbVs can inhibit this effect. Transfections were performed using pRK5-HA-Ub.WT and different combinations of pcDNA3-V5-IBV-PLpro Beaudette, pcDNA3-V5-IBV-PLpro-Beaudette-C, pcDNA3-V5-IBV-PLpro-M41, pcDNA3-V5-IBV-PLpro-M41-C and pcDNA3-FLAG-UbVs. pcDNA3.1-empty vector was used to normalize the total amount of transfected DNA per well. pcDNA3-GFP was co-transfected as a transfection efficiency control. PEI was used as a transfection method. Briefly, per 2 μg of combined plasmid DNA, 6 μL of PEI was diluted in Opti-MEM medium. The plasmid DNA was also diluted in Opti-MEM. Diluted PEI was then added to the diluted DNA, vortexed and incubated for 20 min. The mixture was then added to the cells (12-well plate, 1 × 10^5^ cells/well) in a dropwise manner, and the cells were incubated at 37 °C, 5% CO_2_, for 24 h. At 24 hpt, the cells were lysed in 2× Laemmli Sample Buffer (LSB, 250 mM Tris-base (pH 6.8), 4% SDS, 20% glycerol, 10 mM DTT, 0.01% Bromophenol blue) and boiled, before Western blotting. All DUB assays were performed independently at least three times.

### 4.8. Western Blot Analysis

To visualize the deubiquitination assays, proteins were separated on an 8% SDS-PAGE gel, after which the gel was blotted onto a 0.2 µm Amersham Hybond P PVDF Western blotting membrane (Cytivia/Merck, Marlborough, MA, USA) and blocked in 1× casein (#C5890, Sigma-Aldrich) for 1 h. The membrane was stained overnight with anti-HA antibody to visualize the ubiquitin smears. The membranes were then washed 3 times with PBS containing 0.05% Tween-20 (PBST), followed by incubation for 1 h with the secondary antibody in 0.5× casein. After washing again with PBST, the membranes were incubated in the dark with the tertiary antibody in 0.5× casein for 1 h. The membranes were washed in PBST and PBS and visualized. The membrane was then re-stained with anti-α-Tubulin or anti-β-actin antibody overnight as a loading control. Similar steps were taken for secondary and tertiary antibody incubations.

### 4.9. Immunofluorescence Assay

DF-1 cells were seeded onto coverslips from sustained culture, transfected with UbVs, infected with IBV Beaudette or IBV M41 for 24 h and subsequently fixed in 3% paraformaldehyde. IFAs were performed as described previously [15]. Briefly, after proper fixing, the coverslips were washed with PBS containing 10 mM glycine. The coverslips were then incubated with PBS containing 0.1% Triton X-100 for 10 min to permeabilize the cells. After this, the coverslips were washed in PBS before incubation with primary antibody in PBS with 5% FCS for 1 h. After incubation, the coverslips were washed again with PBS before incubation with secondary antibody in PBS/5% FCS for 1 h. The coverslips were mounted onto microscope slides with ProLong Glass antifade mounting fluid (Thermo Fisher Scientific/Invitrogen). Results were assessed using Zeiss Axioskop 2 and the AxioVision software (version 4.7) (Carl Zeiss Microscopy, LLC, White Plains, NY, USA).

### 4.10. IBV Infection Experiments

To determine the reduction in viral progeny upon UbV expression, DF-1 cells were seeded into 12-well plates and transfected with the different pcDNA3-FLAG-UbV plasmids when the cell confluency was around 70%, before being infected with IBV. Briefly, per 2 μg of plasmid DNA, 6 μL of PEI was diluted in Opti-MEM medium (Lonza). The plasmid DNA was also diluted in Opti-MEM. Diluted PEI was then added to the diluted DNA, vortexed and incubated for 20 min. The mixture was then added to the cells (12-well plate, 1 × 10^5^ cells/well) in a dropwise manner, and the cells were incubated at 37 °C, 5% CO_2_. After 24 h of transfection, the cells were infected with either IBV Beaudette or IBV M41 at an MOI of 0.5. Supernatants were collected 24 h after infection for viral titration and immunofluorescence microscopy.

To determine IBV Beaudette viral titers, the supernatants were subjected to plaque assay on DF-1 cells. Briefly, 10^−1^ to 10^−6^ dilutions of supernatants were made in PBS/DEAE+2%FCS and added to the appropriate wells of 12-well plates. The plates were then incubated like this for an hour while shaking at 37 °C. Afterwards, the inoculum was removed and replaced with Avicel-containing semi-solid overlay, and the plates were incubated for 2–3 days at 37 °C. After 2–3 days, the cells were fixed with 4% formaldehyde and plaques visualized with crystal violet staining. Corresponding graphs were created using Graphpad Prism. Experiments were performed independently 3 times, and plaque assays were performed in duplicate for each sample with similar results. Work using live virus was performed inside biosafety cabinets at Biosafety Level 2 laboratories at either the Roslin Institute, University of Edinburgh, Scotland, or at the Leiden University Medical Center, The Netherlands.

### 4.11. Expression and Purification of Proteins

*E. coli* BL21 (DE3) cells were transformed with the pETM11 vectors mentioned above and grown in Kanamycin-supplemented LB media at 37 °C. Then, 1 mM of IPTG was added when OD_600_ reached 0.5, and cells were further incubated in a shaker at 18 °C for 16 h. Overnight culture was collected by centrifugation, and the cell pellet was lysed by Emulsiflex with lysis buffer (50 mM Tris-HCl pH 7.5, 200 mM NaCl, 10 mM β-mercaptoethanol). Cell lysates were centrifuged to remove cell debris, and the supernatant was mixed with Ni-NTA resin at 4 °C for 1 h, and then Ni-NTA slurry was loaded onto the open column. Resin was washed with the lysis buffer supplemented with 30 mM Imidazole. Proteins were eluted by buffer (50 mM Tris-HCl pH 7.5, 200 mM NaCl, 250 mM Imidazole, 13.3 mM β-mercaptoethanol). For PLpro, the eluate was mixed with TEV protease to cleave off the His6 tag and incubated at 4 °C for 16 h. After cleavage, the sample was loaded on a Ni-NTA column to capture TEV protease and the remaining His6 tag. Flowthrough was further purified by size exclusion chromatography (Superdex 75) with buffer (50 mM Tris-HCl pH 7.5, 50 mM NaCl, 1 mM DTT). Fractions with corresponding retention volumes were pooled together for biochemical assays. For UbV purification, the eluate from Ni-NTA column purification was directly purified by size exclusion chromatography (Superdex 75) without His-tag cleavage.

### 4.12. In Vitro DUB Activity Assay

PLpro enzymes were diluted in the assay buffer (20 mM Tris-HCl pH 8.0, 150 mM NaCl, 2 mM DTT, 0.05% BSA). DUB kinetics were measured for 1 nM PLpro with a range of Ub-AMC substrate concentration (0 to 64 µM). With 10 µL reaction volume in a 384 well plate, free AMC fluorescence was quantified by a plate reader (SpectraMax iD5, Molecular Devices, San Jose, CA, USA) with an excitation wavelength of 380 nm and an emission wavelength of 460 nm. The concentration of free AMC was estimated by the standard curve of AMC standards. Initial velocity of enzyme activity was estimated by the regression to the exponential equation Y = A (1 − exp(−B * t)) + C. Initial velocities of different substrate concentrations were plotted to estimate k_cat_ and K_M_ for the Michaelis–Menten kinetics (V = V_max_ * [S]/(K_M_ + [S]), where [S] is the initial substrate concentration). To determine the IC_50_ of UbVs, an assay was performed with 1 nM of PLpro and 500 nM of Ub-AMC. Using a range of concentrations (0 to 1.28 µM), UbVs were mixed with PLpro before the addition of Ub-AMC and incubated at 20 °C for 15 min. The enzyme reaction was started by adding Ub-AMC, and the fluorescence of free AMC was measured by SpectraMax iD5 with the same parameters as above.

### 4.13. Model Visualization

The X-ray crystal structure of IBV Beaudette PLpro and wildtype Ub complex (PDB: 5BZ0) was visualized by pymol2 (The PyMOL Molecular Graphics System, Version 2.5.4 Schrödinger, LLC, New York, NY, USA).

## Figures and Tables

**Figure 1 ijms-26-05254-f001:**
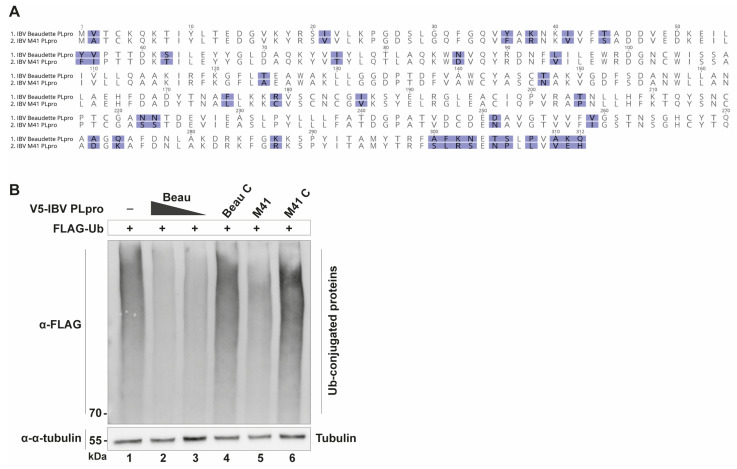
Deubiquitinating activity of IBV PLpro. (**A**) Amino acid alignment of the PLpro enzymes of IBV Beaudette and IBV M41, used to create the mammalian expression vectors. Color shows amino acid sequence diversity. (**B**) DUB activity of IBV Beaudette and M41 PLpro was visualized by co-transfecting DF-1 cells with plasmids encoding wildtype (FLAG-tagged) ubiquitin and (V5-tagged) IBV PLpro from Beaudette (denoted “Beau”, 1600 and 500 ng of plasmid) or M41 (wildtype or their respective catalytic mutant “C”). Lysates were collected 24 hpt and subjected to Western blot analysis.

**Figure 2 ijms-26-05254-f002:**
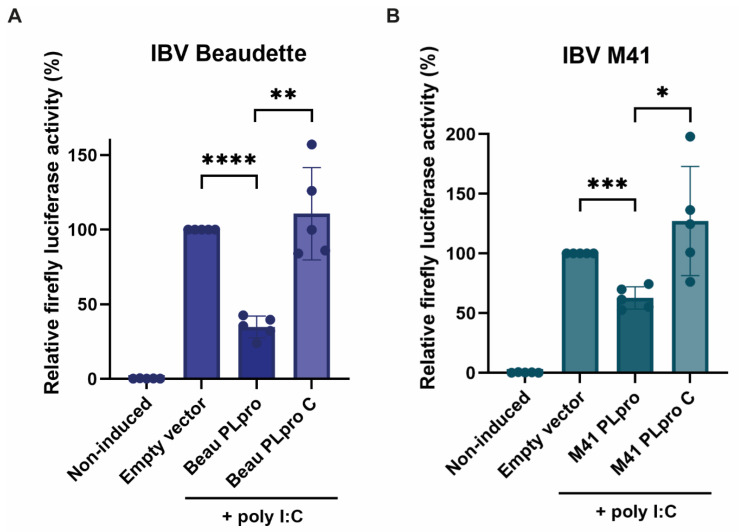
Luciferase-based assay detecting the activity of the chicken IFN-β promoter upon co-expression of PLpro from the IBV Beaudette or M41 strains. Activation of the chIFN-β promoter is represented by the relative firefly luciferase activity, with the empty vector control with poly I:C induction set to 100% activity. Co-transfection of plasmids expressing IBV Beaudette (**A**) or M41 (**B**) PLpro caused a reduction in activity, which was not observed when catalytic mutants of the respective PLpro enzymes (“C”) were expressed instead. Means ± standard deviations are shown for five independent experiments, with four technical replicates in each measurement. Statistical significance is shown for PLpro-transfected conditions relative to empty vector-transfected poly I:C-induced cells and for catalytic mutant-transfected cells relative to the corresponding wildtype PLpro sample (* *p* > 0.05, ** *p* > 0.005, *** *p* > 0.001, **** *p* > 0.0001).

**Figure 3 ijms-26-05254-f003:**
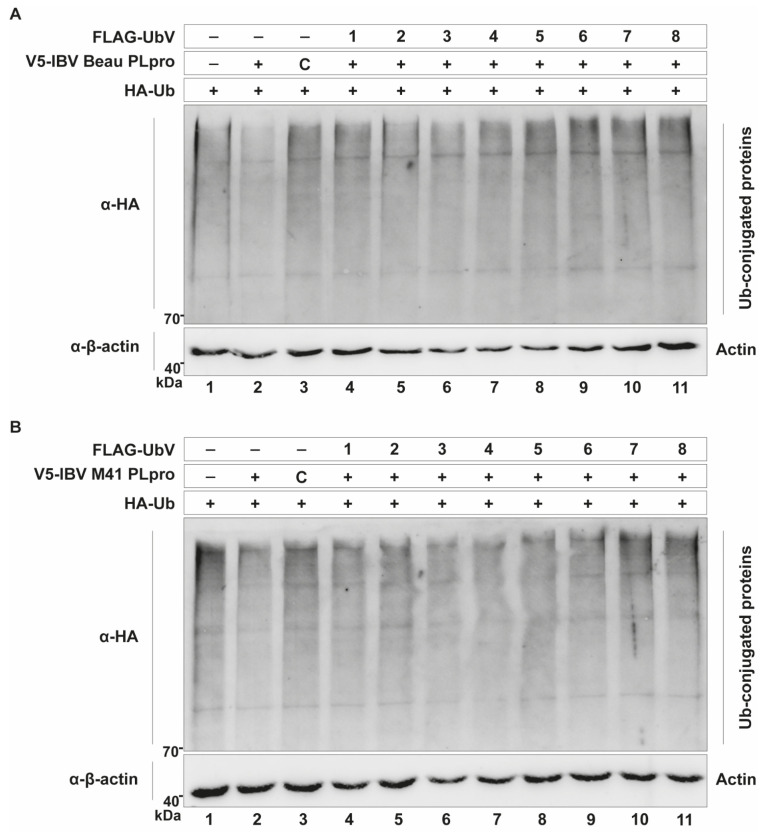
UbVs inhibit the DUB activity of IBV PLpro. DUB activity of IBV Beaudette (**A**) or M41 (**B**) PLpro, observed as a reduction in ubiquitin smear intensity (compare lanes 1 and 2), is inhibited by various UbVs, visualized by the co-transfection of DF-1 cells with constructs expressing wildtype Ub, IBV PLpro (Beaudette or M41) and the different UbVs. Catalytic mutant “C” of both PLpro enzymes was used as the control for complete inhibition of DUB activity. Lysates were harvested 24 hpt and subjected to Western blot analysis.

**Figure 4 ijms-26-05254-f004:**
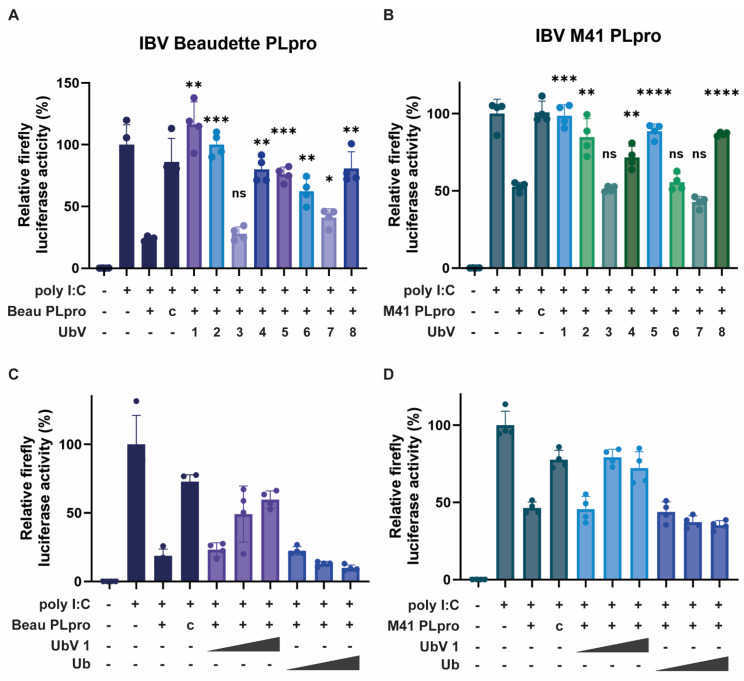
Luciferase assays depicting the efficacy of the UbVs at reducing the inhibition of the chicken IFN-β promoter, caused by IBV PLpro. Activation of the chIFN-β promoter is represented by the relative firefly luciferase activity, with the poly I:C-induced condition set to 100% activity. Co-transfection of plasmids expressing IBV Beaudette (**A**) or M41 (**B**) PLpro caused a reduction in activity, which was not detectable when a plasmid expressing the catalytic mutant (“C”) of the respective PLpro enzymes was co-transfected instead. Co-transfection of the UbVs with IBV Beaudette (**A**) or M41 (**B**) PLpro showed different levels of restoration of the chIFN-β promoter activity. Effects of increasing doses of UbV 1 were tested against IBV Beaudette PLpro (**C**) and IBV M41 PLpro (**D**). Wildtype ubiquitin was used as a negative control. Experiments were performed independently at least three times. Means ± standard deviations are shown for a representative experiment, containing four technical repeats. Statistical significance is shown for UbV co-transfected conditions relative to PLpro-only transfected conditions (* *p* > 0.05, ** *p* > 0.005, *** *p* > 0.001, **** *p* > 0.0001), ns stands for not significant.

**Figure 5 ijms-26-05254-f005:**
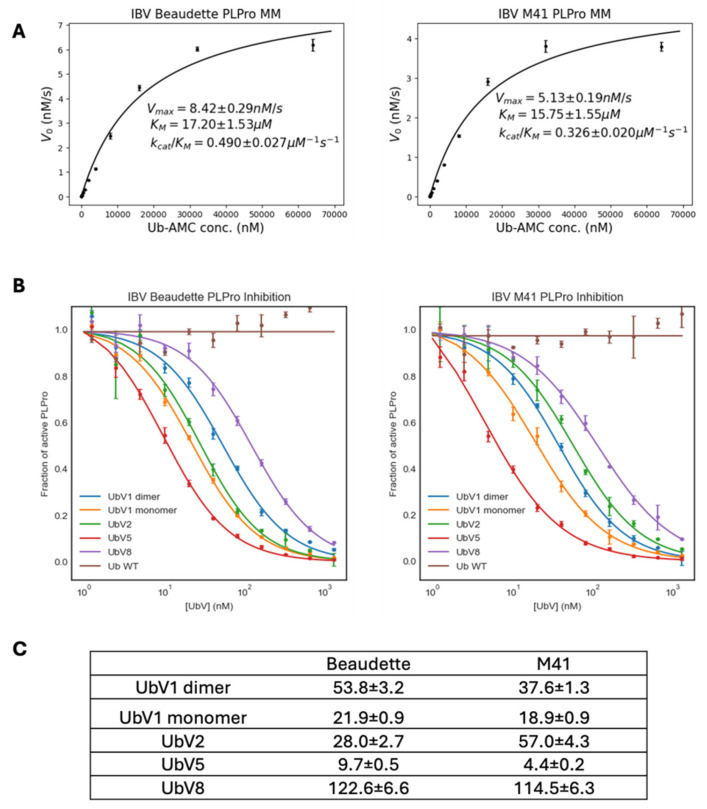
Biochemical characterization of PLpro and IC_50_ measurement of UbVs. Recombinant PLpro enzymes of IBV Beaudette and M41 were purified from an *E. coli* expression system, and the Michaelis–Menten kinetics were measured with Ub-AMC as a substrate (**A**). Inhibition of PLpro DUB activity by UbVs was measured by enzyme inhibition assay (**B**). IC_50_ values, reported here in nM, representing inhibition of PLpro from Beaudette or M41 strains by different UbVs as indicated (**C**). Data points represent the mean of four technical replicates, and the error bars depict standard deviations.

**Figure 6 ijms-26-05254-f006:**
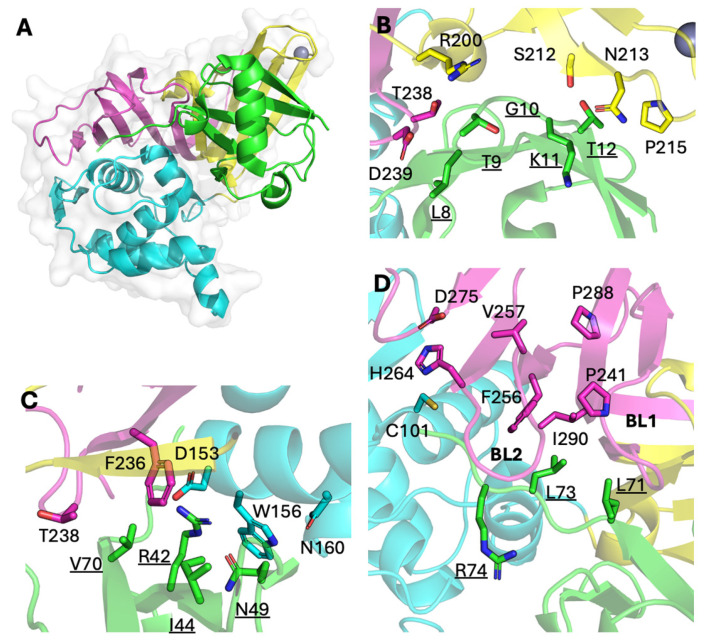
Binding interface between wildtype Ub and Beaudette PLpro. Cartoon representation of wildtype Ub and Beaudette PLpro complex (PDB ID: 5BZ0) (**A**). Wildtype Ub is colored green, the PLpro thumb domain is cyan, the finger domain is yellow, and the palm domain is magenta. Sidechains of key residues in Ub region 1 and its vicinity (**B**). Sidechains of key residues in Ub region 2 and its vicinity (**C**). Sidechains of key residues in Ub region 3 and its vicinity (**D**). Residues of wildtype Ub are underlined.

**Figure 7 ijms-26-05254-f007:**
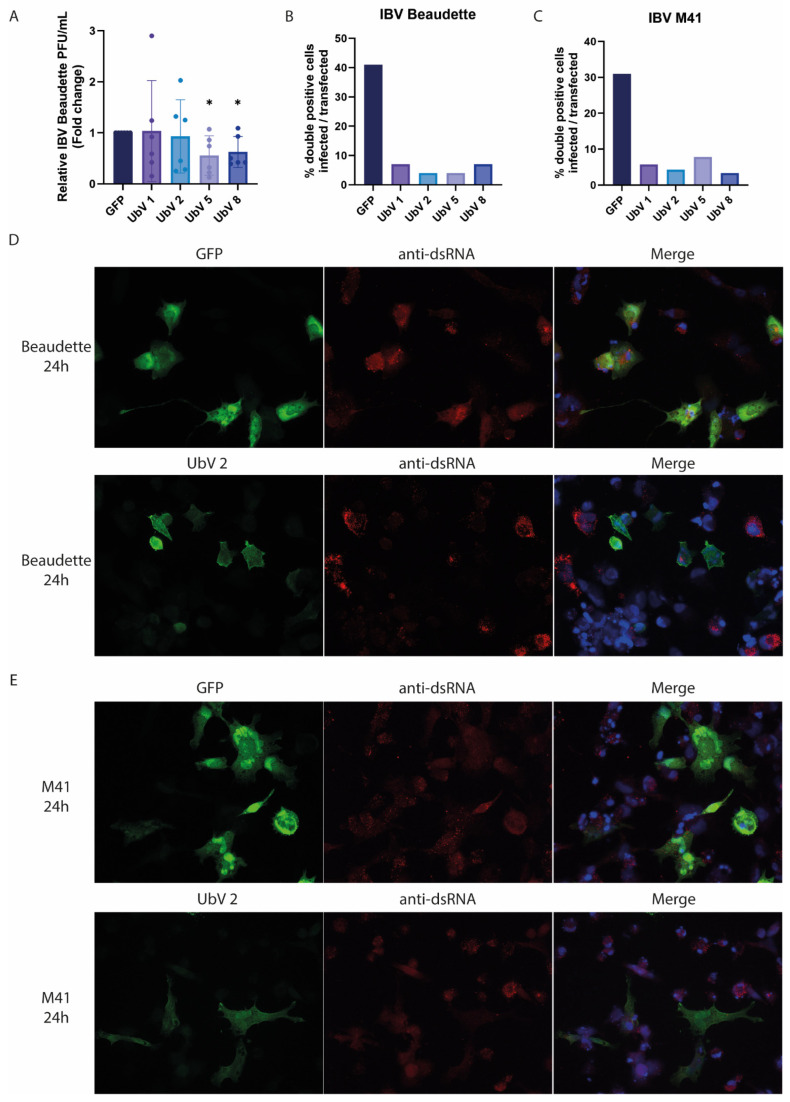
Selected UbVs inhibit IBV infection. (**A**) IBV Beaudette replication was assessed by measuring plaque forming units (PFUs) of virus that were released into the supernatant of the FLAG-UbV- or control-transfected DF-1 cells. Cells were infected with an MOI of 0.5, and IBV titers were subsequently determined for supernatants harvested at 24 hpi. Fold change was calculated relative to GFP control. Means and standard deviations are shown for six independent experiments (* *p* > 0.05). (**B**,**C**) show the percentage of cells that were double positive for GFP/UbV and IBV Beaudette (**B**) or IBV M41 (**C**) infection using replicates of representative IFAs shown in (**D**,**E**), respectively. Immunofluorescence was performed to examine double-stranded RNA (dsRNA) production and thus viral infection in control (GFP) and FLAG-UbV 2-expressing DF-1 cells infected with IBV Beaudette (**D**) or IBV M41 (**E**) and fixed at 24 hpi. Viral staining (dsRNA) is shown in red, GFP and FLAG-UbV are shown in green, and Hoechst is depicted in blue, 40× magnification.

**Table 1 ijms-26-05254-t001:** Engineered UbVs selected by IBV PLpro binding. Overview of amino acid sequence differences comparing wildtype ubiquitin (Ub WT) and UbVs selected by screening with IBV PLpro. Ubiquitin amino acid residue numbers are shown at the top. The table shows only those Ub regions that were subjected to diversification in the phage-displayed UbV library.

	Region 1						Region 2					Region 3										
	2	8	9	10	11	14		42	44	49		62	63	66	70	71	72	74	75	76	77	78
**Ub WT**	**Q**	**L**	**T**	**G**	**K**	**T**		**R**	**I**	**Q**		**Q**	**K**	**T**	**V**	**L**	**R**	**R**	**G**	**G**	**-**	**-**
UbV.IBV.1	-	I	S	A	-	-		G	M	F		P	-	-	L	M	-	-	-	R	S	V
UbV.IBV.2	K	N	V	G	-	S		G	M	P		K	Q	S	L	-	-	-	R	L	A	L
UbV.IBV.3	-	-	-	G	-	-		G	M	-		-	-	-	-	-	-	-	-	-	G	G
UbV.IBV.4	-	-	-	G	-	-		G	M	-		-	-	-	-	-	-	-	-	-	-	-
UbV.IBV.5	-	F	-	R	N	-		-	-	Y		L	-	-	L	-	N	-	-	L	F	N
UbV.IBV.6	-	F	-	R	N	-		-	-	-		-	-	-	-	-	-	-	-	-	G	G
UbV.IBV.7	-	F	-	R	N	-		-	-	-		-	-	-	-	-	-	-	-	-	-	-
UbV.IBV.8	L	F	-	R	N	A		-	-	-		-	N	W	L	R	S	L	S	P	T	H

## Data Availability

The original contributions presented in this study are included in the article/Appendix A. Further inquiries can be directed to the corresponding author(s).

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
