# Peer review of "Inhibiting Infectious Bronchitis Virus PLpro Using Ubiquitin Variants"

_ijms, 2025, doi:10.3390/ijms26115254_

Round 1

Reviewer 1 Report

Comments and Suggestions for Authors

Comments and Suggestions
Summary Section
1. Include other epidemiological aspects of the virus, such as bird mortality and epidemic potential.
2. State the conclusions of the study in quantitative terms.

Introduction Section
3. Mention that other species can be infected by this virus.
4. Mention key aspects of viral genetics related to your research.
Results Section
5. Avoid redundant information between the text and graphics.
Methodology Section:
6. Improve the paragraph on deubiquitination assays.
Conclusion Section:
7. Clarify the conclusions of this study.

Author Response

We thank the reviewers very much for their comments. Below we have copied the reports for reference and will address the comments point-by-point. When the manuscript was changed and line numbers are mentioned, these refer to the revised manuscript with tracked changes.

Comments Reviewer #1:
Summary Section

Comment 1: Include other epidemiological aspects of the virus, such as bird mortality and epidemic potential.

Response 1: We thank the reviewer for their suggestion. We have now added to the abstract that IBV is a global problem, which affects egg production and mortality upon secondary bacterial infections. Please find the edits in lines 14-16 of the revised manuscript.

Comment 2: State the conclusions of the study in quantitative terms.

Response 2: We think this is a useful suggestion. Therefore, we have edited line 24-28 in the manuscript to state that the results described are based on immunofluorescence microscopy and have described the results in a more quantitative fashion.

Introduction Section

Comment 3: Mention that other species can be infected by this virus.

Response 3: We have adjusted the text of the introduction to include that IBV can infect other avian species besides chickens at lines 40-41. We have also included an appropriate reference.

Comment 4: Mention key aspects of viral genetics related to your research.

Response 4: In this manuscript we investigate the possibility of using UbVs to inhibit the viral protease PLpro. A description of PLpro and its functionalities with respect to viral genome processing is described in lines 53-67. Furthermore, this is a proof-of-concept study, investigating the efficacy of ubiquitin analogues in the context of the Beaudette and M41 strains. Broad applicability across all of the IBV strains would go beyond the scope of this study.

Results Section

Comment 5: Avoid redundant information between the text and graphics. 

Response 5: We think that figures should be fully understandable from the legends alone without referral to the text. This may mean that there are some duplications between figure legends and text, but we would like to maintain this for the sake of clarity, as explained.

Methodology Section

Comment 6: Improve the paragraph on deubiquitination assays.

Response 6: In order to further clarify the methodology of the deubiquitination assays we have added an introductory sentence to explain the purpose of the assay and also provided a reference to a previously published paper from our group that describe this assay, so that readers can refer to this if needed. Lines 622-626.

Conclusion Section

Comment 7: Clarify the conclusions of this study.

Response 7: We have reviewed the text about the conclusions of the study (from line 462), but feel that this clearly explains the results and places the conclusions in the context of previous and future research. We have further clarified, as suggested by the reviewer, the quantitative reduction of viral infection observed in the fluorescence infection assays in the abstract/summary section. For the rest, we kept the text as it was.

Reviewer 2 Report

Comments and Suggestions for Authors

Papain-like protease (PLpro) of coronavirus (CoV) performs proteolytic maturation of non-structural proteins that play a role in virus replication and performs deubiquitination of host cell factors to hinder antiviral responses. The avian infectious bronchitis virus (IBV), the causative agent of bronchitis in chickens that causes huge economic losses in the poultry industry globally each year, encodes a PLpro. The substrate specificities of this PLpro are not clearly understood.The authors carry out a series of experimental tests including microscopy to prove the antiviral activity of their work and also use immunohistochemical and immuno microscopic techniques. The work is well structured. I ask the authors only to make a brief description of the system PLpro in the introduction.

Their data on UVB’s  thah they have been found to inhibit the deubiquitinating activity of PLpro, are considerable.

Author Response

We thank the reviewers very much for their comments. Below we have copied the reports for reference and will address the comments point-by-point. When the manuscript was changed and line numbers are mentioned, these refer to the revised manuscript with tracked changes.

Comments Reviewer #2:

Papain-like protease (PLpro) of coronavirus (CoV) performs proteolytic maturation of non-structural proteins that play a role in virus replication and performs deubiquitination of host cell factors to hinder antiviral responses. The avian infectious bronchitis virus (IBV), the causative agent of bronchitis in chickens that causes huge economic losses in the poultry industry globally each year, encodes a PLpro. The substrate specificities of this PLpro are not clearly understood. The authors carry out a series of experimental tests including microscopy to prove the antiviral activity of their work and also use immunohistochemical and immuno microscopic techniques. The work is well structured. I ask the authors only to make a brief description of the system PLpro in the introduction.

Their data on UVB’s  that they have been found to inhibit the deubiquitinating activity of PLpro, are considerable.

Response: We thank the reviewer for these positive comments. In our manuscript we investigate the possibility of using UbVs to inhibit the viral protease PLpro. We do not understand what the reviewer means by “the system PLpro”, but as we have clearly introduced PLpro as one of the proteases encoded by coronaviruses in the introduction, we have kept the text as it is.

Reviewer 3 Report

Comments and Suggestions for Authors

In the manuscript entitles” Inhibiting infectious bronchitis virus PLpro using ubiquitin variants” the authors developed ubiquitin variants to inhibit IBV infection by blocking proteolytic activity of PLpro. The manuscript is addressed an interesting topic and well- written. I have minor comments/edits.

Line 114: Figure 1. B why it includes 2 lanes for Beau? Are they different? please explain in figure legend.

Material and methods: please add the statistical analysis used? GraphPad Prism is mentioned but what are tests used for comparison/ significance.

Line 710: Author Contributions, please revise adding authors initials.

Line 732: References, revise references following the journal guidelines.

Author Response

We thank the reviewers very much for their comments. Below we have copied the reports for reference and will address the comments point-by-point. When the manuscript was changed and line numbers are mentioned, these refer to the revised manuscript with tracked changes.

Comments Reviewer #3:

In the manuscript entitled “Inhibiting infectious bronchitis virus PLpro using ubiquitin variants” the authors developed ubiquitin variants to inhibit IBV infection by blocking proteolytic activity of PLpro. The manuscript addressed an interesting topic and is well- written. I have minor comments/edits.

Comment 1: Line 114: Figure 1. B why it includes 2 lanes for Beau? Are they different? please explain in figure legend.

Response 1: We think this is a very important suggestion. We have adjusted the figure legend to clarify that two different amounts of the plasmid encoding for Beau PLpro were tested, a higher and a lower amount. In the revised version of the manuscript this now refers to line 125.

Comment 2: Material and methods: please add the statistical analysis used? GraphPad Prism is mentioned but what are tests used for comparison/ significance.

Response 2: We thank the reviewer for this comment, this important information was indeed missing. We have now clarified in line 616 that unpaired t-tests were applied to determine significance.

Comment 3: Line 710: Author Contributions, please revise adding authors initials.

Response 3: This information has now been added. It had already been filled in in the submission system, but was not added  to the manuscript, which was an oversight for which we apologize.

Comment 4: Line 732: References, revise references following the journal guidelines.

Response 4: The references have now been formatted according to the journal guidelines using the reference style made available for our reference manager.